# Sustained *Helicobacter pylori* infection accelerates gastric dysplasia in a mouse model

Valerie P O'Brien[1], Amanda L Koehne[2,3], Julien Dubrulle[4], Armando E Rodriguez[1], Christina K Leverich[1], V Paul Kong[3], Jean S Campbell[5], Robert H Pierce[5], James R Goldenring[6,7], Eunyoung Choi[6], Nina R Salama[1]

More than 80% of gastric cancer is attributable to stomach infection with *Helicobacter pylori* (*Hp*). Gastric preoplastic progression involves sequential tissue changes, including loss of parietal cells, metaplasia and dysplasia. In transgenic mice, active KRAS expression recapitulates these tissue changes in the absence of *Hp* infection. This model provides an experimental system to investigate additional roles of *Hp* in preneoplastic progression, beyond its known role in initiating inflammation. Tissue histology, gene expression, the immune cell repertoire, and metaplasia and dysplasia marker expression were assessed in KRAS+ mice +/−*Hp* infection. *Hp*+/KRAS+ mice had severe T-cell infiltration and altered macrophage polarization; a different trajectory of metaplasia; more dysplastic glands; and greater proliferation of metaplastic and dysplastic glands. Eradication of *Hp* with antibiotics, even after onset of metaplasia, prevented or reversed these tissue phenotypes. These results suggest that gastric preneoplastic progression differs between *Hp*+ and *Hp*− cases, and that sustained *Hp* infection can promote the later stages of gastric preneoplastic progression.

## Introduction

About 13% of the global cancer burden in 2018 was attributable to carcinogenic infections (de Martel et al, 2020), and *Helicobacter pylori* (*Hp*)–associated gastric cancer accounted for the largest proportion of these cancers (Plummer et al, 2015). More than 77% of new gastric cancer cases, and more than 89% of new non-cardiac gastric cancer cases, were attributable to infection with *Hp* (de Martel et al, 2020), a bacterium that colonizes the stomach of half the world's population (Suerbaum et al, 2002). However, *Hp* infection confers only a 1–2% lifetime risk of developing stomach

cancer (Kuipers, 1999) and thus a complex interplay between the bacterium and host is presumed to lead to cancer development in only some individuals.

The exact mechanisms through which *Hp* infection promotes gastric cancer remain largely elusive. *Hp* infection typically occurs during childhood and always causes chronic inflammation (gastritis) (Kusters et al, 2006). *Hp*-dependent chronic inflammation promotes the accumulation of reactive oxygen species and other toxic products that cause mutations in gastric epithelial cells (Chaturvedi et al, 2004, 2011; Allen et al, 2005). Early studies using tissue histology rarely detected *Hp* in tumors, leading to a belief that *Hp* triggers the initial inflammatory insult in the stomach, but that *Hp* is essentially irrelevant by the time gastric cancer is detected; in other words, once chronic gastric inflammation develops and oncogenic pathways are activated, the presence of *Hp* is no longer necessary to promote metaplastic changes that lead to cancer. However, more sensitive molecular methods detect *Hp* in about half of tumors (Tang et al, 2005; Cristescu et al, 2015; Talarico et al, 2018), and eradication of *Hp* combined with tumor resection helps prevent tumor recurrence (Choi et al, 2018), suggesting that *Hp* may promote the later stages of metaplasia and cancer development in at least some individuals.

Beyond eliciting oncogenic mutations, the mechanism(s) through which chronic gastritis might promote gastric cancer development is not well understood (Salama et al, 2013). Humans generally develop a strong $T_h1$ and $T_h17$ immune response against *Hp* that helps control the infection (Akhiani et al, 2002; Sayi et al, 2009; Velin et al, 2009). This T-cell response does not clear the infection and furthermore can drive immunopathology in the gastric mucosa (Stoicov et al, 2009; Shi et al, 2010), and *Hp* infection can disrupt normal T-cell function through multiple mechanisms (Gebert et al, 2003; Das et al, 2006; Salama et al, 2013). Thus, T cells can play both protective and detrimental roles during *Hp* stomach infection. More broadly, anticancer immunity in the context of gastric cancer is not well understood. A better understanding of how active *Hp* infection may impact

---

[1]Fred Hutchinson Cancer Research Center, Human Biology Division, Seattle, WA, USA   [2]Fred Hutchinson Cancer Research Center, Comparative Medicine Shared Resource, Seattle, WA, USA   [3]Fred Hutchinson Cancer Research Center, Experimental Histopathology Shared Resource, Seattle, WA, USA   [4]Fred Hutchinson Cancer Research Center, Genomics and Bioinformatics Shared Resource, Seattle, WA, USA   [5]Fred Hutchinson Cancer Research Center, Program in Immunology, Seattle, WA, USA   [6]Department of Surgery, Epithelial Biology Center, Department of Cell and Developmental Biology, Vanderbilt University School of Medicine, Nashville, TN, USA   [7]Nashville Veterans Affairs Medical Center, Nashville, TN, USA

Correspondence: nsalama@fredhutch.org

gastric inflammation in the context of metaplasia and cancer development may lead to the discovery of new drug targets or therapeutic strategies.

The *Mist1-Kras* mouse is one of the only existing mouse models to recapitulate the progression from healthy gastric epithelium to spasmolytic polypeptide-expressing metaplasia (SPEM), intestinal metaplasia (IM), and dysplasia (Choi et al, 2016). This model uses KRAS, a GTPase signaling protein of the Ras (Rat Sarcoma) family that regulates cell survival, proliferation, and differentiation (Campbell et al, 1998; Jackson et al, 2001). Molecular profiling studies have shown that about 40% of gastric tumors have signatures of RAS activity (Deng et al, 2012; Cancer Genome Atlas Research Network, 2014). In the mouse model, treatment with tamoxifen (TMX) induces the expression of a constitutively active *Kras* allele (G12D) in the gastric chief cells. Within 1 mo, SPEM develops in 95% of corpus glands, and over the next 3 mo progresses to IM (Choi et al, 2016). Thus, active KRAS expression in mice serves as a tool to recapitulate changes that, in humans, are induced by years of inflammation due to *Hp* infection. We used *Mist1-Kras* mice to test our hypothesis that *Hp*, if present during metaplasia and dysplasia, could impact pathology. We found that, counter to the belief that *Hp* is only important for initiating inflammation, sustained *Hp* infection coupled with active KRAS expression led to severe inflammation, altered metaplasia marker expression, and increased cell proliferation and dysplasia compared with *Hp*–/KRAS+ mice. Thus, the course of gastric neoplastic progression may differ depending on whether *Hp* is present during the later stages of disease progression.

## Results

### *Hp* infection worsens gastric immunopathology in mice expressing active KRAS

To assess whether *Hp* impacts KRAS-driven metaplasia, we performed concomitant infection/induction experiments in *Mist1-Kras* mice. First mice were infected with *Hp*, or mock-infected, and the next day the mice were treated with tamoxifen (TMX) to induce active KRAS expression in stomach chief cells, or sham-induced. After 2, 6, or 12 wk, mice were humanely euthanized and stomachs were aseptically harvested and used for downstream analyses (Fig 1). Formalin-fixed, paraffin-embedded (FFPE) tissue sections were used for histological analysis of the corpus (Fig 2), where active KRAS is expressed in TMX-induced *Mist1-Kras* mice. Compared with *Hp*–/KRAS– mice (Fig 2A and B), *Hp* infection alone caused modest inflammation at 2 wk that increased over time, with loss of parietal cells by 6 wk and moderate surface epithelial hyperplasia by 12 wk (Fig 2C and D). Mice expressing active KRAS had far more striking changes to the tissue over time (Fig 2E–H). To quantify the effects of *Hp* infection in this model, a blinded analysis was performed to assess inflammation, oxyntic atrophy (loss of parietal cells), and surface epithelial hyperplasia in active KRAS-expressing mice (Fig 3A–C).

KRAS expression caused changes to the corpus epithelium that were apparent within 2 wk, with a moderate degree of inflammation,

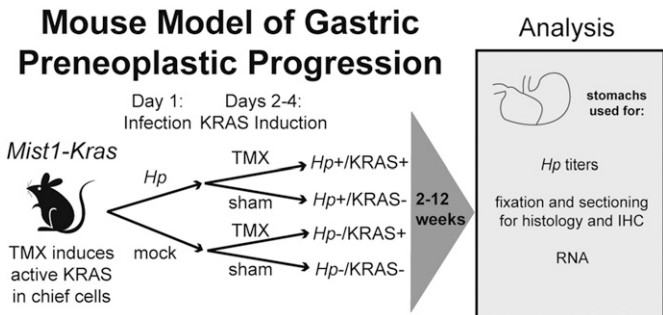

**Figure 1. *Mist1-Kras* mice were used to assess whether and how *Hp* infection alters gastric preneoplastic progression.**
On day one, mice are infected with *H. pylori* (*Hp*) by oral gavage, or mock-infected. On days two through four, mice receive daily injections with tamoxifen (TMX) to induce a constitutively active *Kras* allele (G12D) in the chief cells (*Mist1*-expressing) of the stomach. After 2, 6, or 12 wk, mice are humanely euthanized and the glandular stomach (excluding forestomach region) is assessed as indicated.

surface epithelial hyperplasia, and some loss of parietal cells. These changes were slightly more severe in a subset of *Hp*+/KRAS+ mice, but overall there were no significant histological differences between *Hp*–/KRAS+ and *Hp*+/KRAS+ mice at this early time point (Fig 2E and G). By 6 wk, each of these parameters became more severe, and notably, parietal cell loss was significantly greater in *Hp*+/KRAS+ mice compared with *Hp*–/KRAS+ mice (Fig 3A). By 12 wk, *Hp*–/KRAS+ mice had mucinous cells, in line with previous observations (Choi et al, 2016) (Fig 2F). *Hp*+/KRAS+ mice looked different from *Hp*–/KRAS+ mice, with loss of normal basal polarity of epithelial cells, and gland architecture that was severely disrupted, including forked or star-shaped gland structure indicative of extensive branching and disorganized maturation (Fig 2G). As well, these mice had hyperchromatic nuclei with variations in nuclear size, which can indicate dysplasia. Moreover, at 12 wk *Hp*+/KRAS+ mice had significantly increased inflammation, parietal cell loss and surface epithelial hyperplasia compared with *Hp*–/KRAS+ mice (Fig 3A–C). Finally, the overall histology score (histological activity index [HAI]), which sums the above scores along with scores for other parameters like epithelial defects and hyalinosis (Fig S1) and which thus indicates the degree of overall immunopathology (Rogers, 2012), was significantly increased in *Hp*+/KRAS+ mice compared with *Hp*–/KRAS+ mice at both 6 and 12 wk (Fig 3D). Thus, concomitant *Hp* infection and active KRAS expression in the corpus leads to histopathological changes to the tissue within 6 wk that become more severe by 12 wk.

The striking inflammation seen in *Hp*+/KRAS+ mice compared with *Hp*+/KRAS– mice might be expected to eliminate *Hp* infection. However, *Hp* was recovered from most KRAS+ mice by stomach culturing (Fig 3E), demonstrating that the bacterium could to some extent withstand the severe inflammation of the preneoplastic stomach. At 2 wk, *Hp* titers did not differ between *Hp*+/KRAS–and *Hp*+/KRAS+ mice, suggesting that the early histopathological changes did not impact bacterial colonization. In sham-induced (KRAS–) mice, *Hp* titers were similar at 6 and 12 wk, and in both cases were lower than at 2 wk, likely because of the onset of adaptive immunity to control the infection. However, in *Hp*+/KRAS+ mice, the contraction of

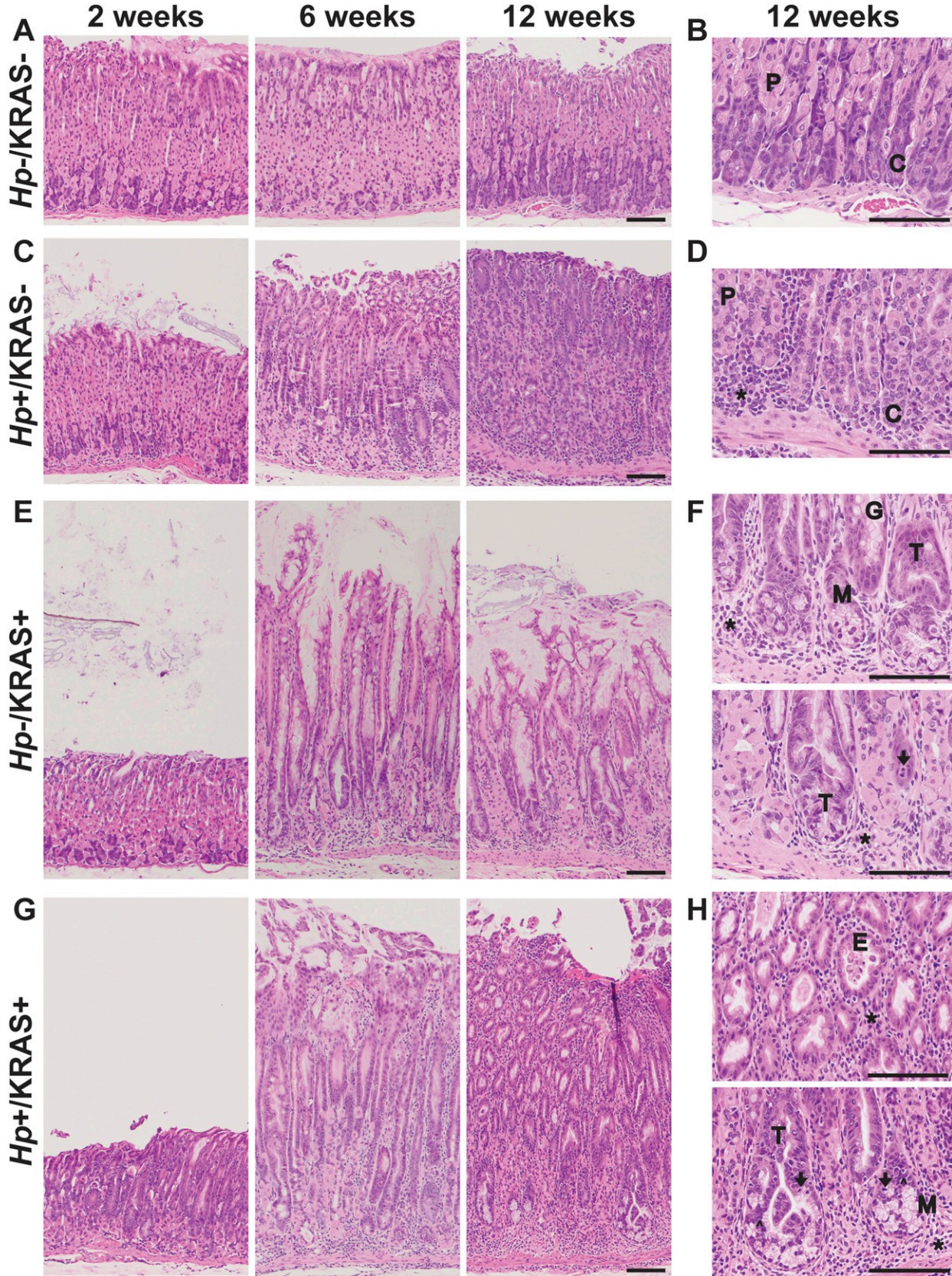

**Figure 2. Concomitant *Hp* infection and active KRAS expression changes tissue histology.**
Formalin-fixed, paraffin-embedded corpus tissue was stained with hematoxylin & eosin and examined with a Nikon Eclipse 50i microscope using 10× and 40× objectives. Images were taken with an Olympus DP27 camera using cellSens 2.3 acquisition software (Olympus). **(A, B, C, D, E, F, G, H)** Shown are representative images from *Hp*−/KRAS− (A, B), *Hp*+/KRAS− (C, D), *Hp*−/KRAS+ (E, F), and *Hp*+/KRAS+ (G, H) mice (n = 10–16 per group) after 2, 6, or 12 wk as indicated. Scale bars, 100 μm. **(B, D, F, H)**

the *Hp* population was greater, with *Hp* recovered from only 10 out of 12 animals at 6 wk and 10 out of 13 animals at 12 wk. *Hp* could be detected within glands by immunofluorescence microscopy (Fig S2). No differences in titer or overall histology score were observed between male and female mice. Interestingly, stomach *Hp* loads were not correlated with histology scores (Fig 3F). We therefore hypothesized that the host inflammatory response to *Hp* infection might contribute to *Hp*-dependent tissue changes.

### *Hp* infection increases and alters KRAS-driven inflammation

Activated KRAS expression itself elicits inflammation: by 12 wk, the median corpus inflammation score in *Hp*–/KRAS+ mice was two (Fig 3B), denoting coalescing aggregates of inflammatory cells in the submucosa and/or mucosa (Rogers, 2012). In this histological evaluation, the addition of *Hp* significantly increased inflammation by 12 wk (*P* < 0.01), with the median score rising to three (denoting organized immune cell nodules in the submucosa and/or mucosa). To characterize the nature of the inflammation, gene expression changes at 12 wk were assessed with a NanoString Mouse Immunology panel (Fig 4 and Table S1). Transcripts of *Cd45*, a pan-immune cell marker, were significantly increased in *Hp*+/KRAS– mice and especially in *Hp*+/KRAS+ mice compared with *Hp*– mice (Fig S3), suggesting greater numbers of immune cells in infected mice. Of the 561 genes in the panel, 60 had no detectable expression among any mice (n = 25) and were excluded from subsequent analysis. Hierarchical clustering was performed on the remaining 501 genes and revealed distinct clustering of the treatment groups (Fig 4A). All *Hp*+ mice clustered separately from *Hp*– mice, demonstrating that infection had the greatest impact on gene expression. Within both the *Hp*+ and the *Hp*– cluster, KRAS+ mice clustered separately from KRAS– mice, suggesting that active KRAS expression also impacted inflammatory gene expression, although to a lesser extent than *Hp* infection did.

Next we assessed gene expression patterns in the different mouse groups. Compared with *Hp*–/KRAS+ mice, *Hp*+/KRAS+ mice had 235 significantly differentially expressed genes ($P_{adjusted}$ < 0.05) (Fig 4B). Several of the most highly up-regulated genes, including *Cd3d*, *Cd3e*, *Cd4*, *Cd8a*, *Gzma*, *Ctla4*, *Icos*, and *Cd6*, implicated a strong T-cell response, in accordance with previous studies in humans and naive animal models (Eaton et al, 2001; Ernst et al, 1997). Likewise, compared with *Hp*–/KRAS– mice, *Hp*+/KRAS– mice had 177 differentially expressed genes, with *Cd3d*, *Cd3e*, *Cd4*, *Cd8a*, *Gzma*, *Ctla4*, *Icos*, and *Cd6* once again highly significantly differentially expressed (Fig 4C). Thus, many of the gene expression differences seen in *Hp*+/KRAS+ mice versus *Hp*–/KRAS+ mice are likely reflective of a general pattern of *Hp*-mediated inflammation that is independent of the metaplastic state of the tissue. However, we identified a unique inflammatory gene signature in *Hp*+/KRAS+ mice (Fig 4D), demonstrating that the inflammation observed in this group is not only of a greater magnitude than in the other groups but also of a different nature. We identified 46 genes whose expression (normalized to *Hp*–/KRAS– mice) was >2-fold increased or

decreased in *Hp*+/KRAS+ mice, but <1.5-fold increased or decreased in *Hp*+/KRAS– and *Hp*–/KRAS+ mice. Many of these genes implicated T cells (*Ccr6*, *Cd27*, *Cd53*, *Cxcl11*, *Foxp3*, *Gata3*, *Il12b*, *Pdcd1lg2* [PD-L2], *Tigit*, and *Tnfsf18* up-regulated; *Il17re* down-regulated) and macrophages (*Ccl3*, *Csf1r*, *Emr1* [F4/80], *Il1a*, and *Irf5* up-regulated). As well, most markers of T-cell exhaustion (Blackburn et al, 2009; Saeidi et al, 2018) were only strongly expressed in *Hp*+/KRAS+ mice (Fig S4). Thus, even though both *Hp*+/KRAS+ mice and *Hp*+/KRAS– mice had significant up-regulation of T-cell–related genes compared with their *Hp*– counterparts, the addition of active KRAS may impact the nature of T-cell polarization and function.

In animal models, immune pressure due to chronic *Hp* infection results in loss of function of the *Hp* type IV secretion system (T4SS) (Barrozo et al, 2013). *Hp* strains isolated from long-term experimental infections of C57BL/6 mice (but not *Rag1* mice deficient in adaptive immune responses), gerbils and monkeys lose their ability to elicit IL-8 secretion by gastric epithelial cells in vitro (Barrozo et al, 2013). In line with these observations, we found that ~50% of *Hp* strains isolated from 12+ wk infections of KRAS– mice had lost their T4SS activity (Fig S5). Surprisingly, *Hp* strains isolated from KRAS+ mice were no more likely to lose their T4SS activity, despite the severe inflammation seen in these animals.

### *Hp*+/KRAS+ mice have T cells throughout the lamina propria and fewer M2 macrophages

To detect immune cell subsets in the corpus of *Hp*–/KRAS+ versus *Hp*+/KRAS+ mice at 12 wk, we performed multiplex fluorescent immunohistochemistry (IHC) with the following markers: for T cells, CD3, CD4, CD8α, FOXP3 (regulatory T cell marker), and PD-1 (T-cell exhaustion marker); for macrophages, F4/80 and the polarization markers MHC class II (M1 macrophages) and CD163 (M2 macrophages) (Fig 5). HALO software was used to detect and enumerate immune cell subsets (Fig S6). In *Hp*–/KRAS+ mice we detected moderate numbers of CD3+ T cells, most of which were CD4+, and a few of which were CD8α+ (Figs 5A and S6A). *Hp*+/KRAS+ mice had significantly more CD3+ T cells, but the proportion of CD4+ versus CD8α+ cells was similar, with more CD4+ than CD8α+ cells. Interestingly, most of the CD3+ cells in *Hp*–/KRAS+ mice expressed FOXP3 and PD-1 (Fig 5B), suggesting they may be activated regulatory T cells (Lowther et al, 2016). In *Hp*+/KRAS+ mice, there were significantly more FOXP3+ cells (Fig S6A), some of which were PD-1 double positive (Fig 5B). However, many CD3+ cells did not express either of these markers, suggesting they may be different T-cell subsets than are found in *Hp*–/KRAS+ mice, and/or NK cells. Cell localization was also different between treatment groups: in *Hp*–/KRAS+ mice, most T cells were located at the base of the glands, whereas in *Hp*+/KRAS+ mice, T cells were located throughout the glands. Finally, both groups of mice had F4/80+ cells throughout the lamina propria (Figs 5C and S6B), suggesting presence of macrophages or eosinophils (McGarry & Stewart, 1991). We previously found that M2 macrophages promoted SPEM progression in mice and were associated with human SPEM and IM (Petersen et al, 2014). In *Hp*–/KRAS+ mice, some F4/80+ cells were dual-positive for the M2 polarization marker CD163, in line with previous findings (Choi et al, 2016), and some were dual-positive for

Examples of the following morphological features are designated in the higher magnification images in (B, D, F, H): chief cells (C); glandular ectasia (E); goblet-like cell morphology (G); hyperchromatic nuclei (carets); immune cells (asterisks); mitotic figures (arrows); mucus (M); parietal cells (P); tortuous glands (T).

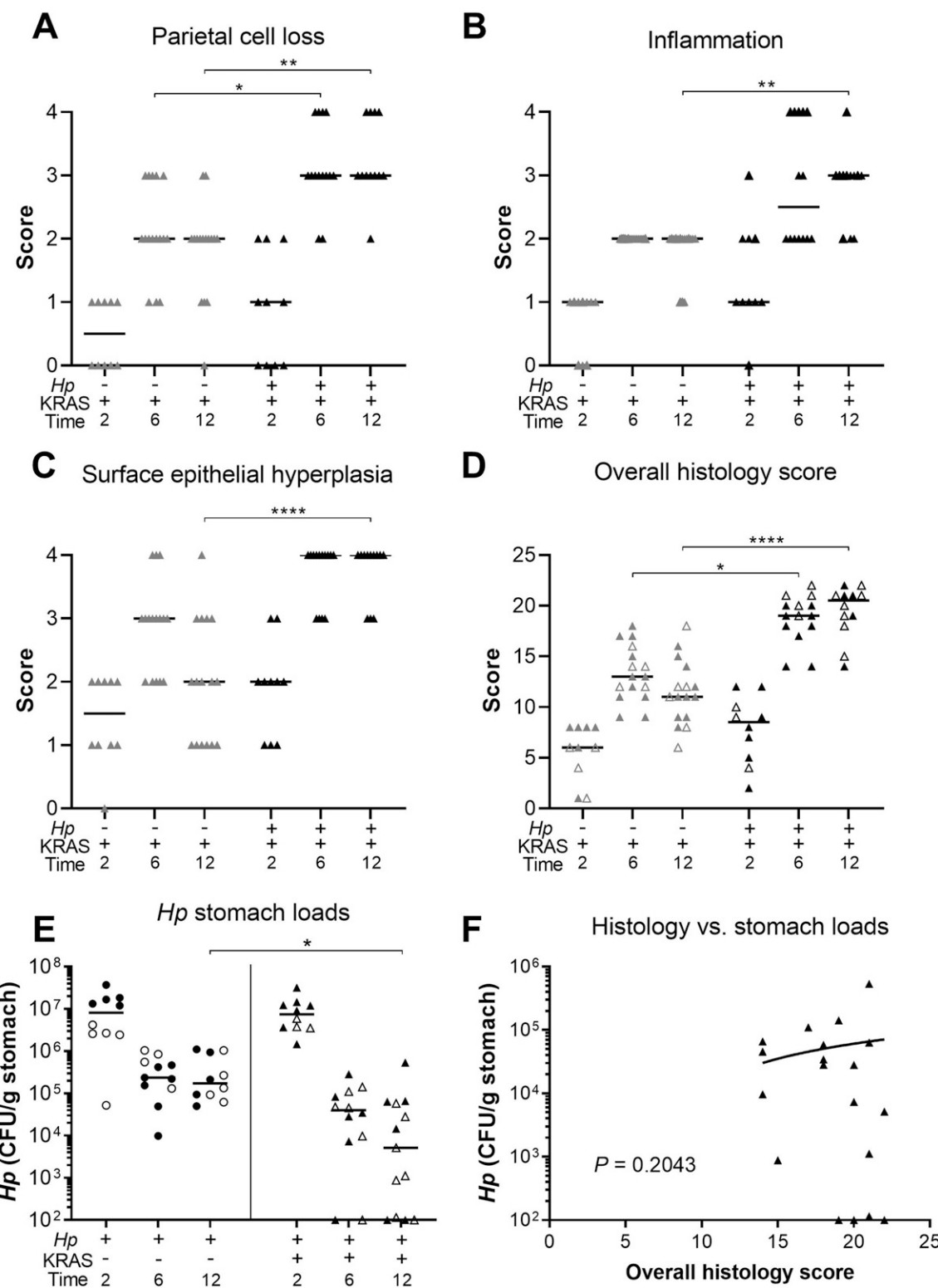

**Figure 3. *Hp*+/KRAS+ mice have severe gastric immunopathology marked by inflammation, loss of parietal cells, and surface epithelial hyperplasia.**
Stomachs from n = 10–16 mice per group were evaluated for tissue pathology and bacterial colonization. **(A, B, C, D)** Hematoxylin-and-eosin–stained corpus tissue was assessed for parietal cell loss (oxyntic atrophy) (A), inflammation (B), surface epithelial hyperplasia (C), and the composite histological activity index (D) in a blinded fashion using the Rogers criteria (Rogers, 2012). **(E)** *Hp* loads were assessed by quantitative culture; mice with no detectable colonization were plotted at the limit of detection. **(F)** Comparison of *Hp* loads and overall histology score (from D) from the same mouse at 6 or 12 wk. Significance was assessed by Spearman correlation. Data are combined from N = 2 independent mouse experiments per time point. Data points represent actual values for each individual mouse and bars indicate median values.

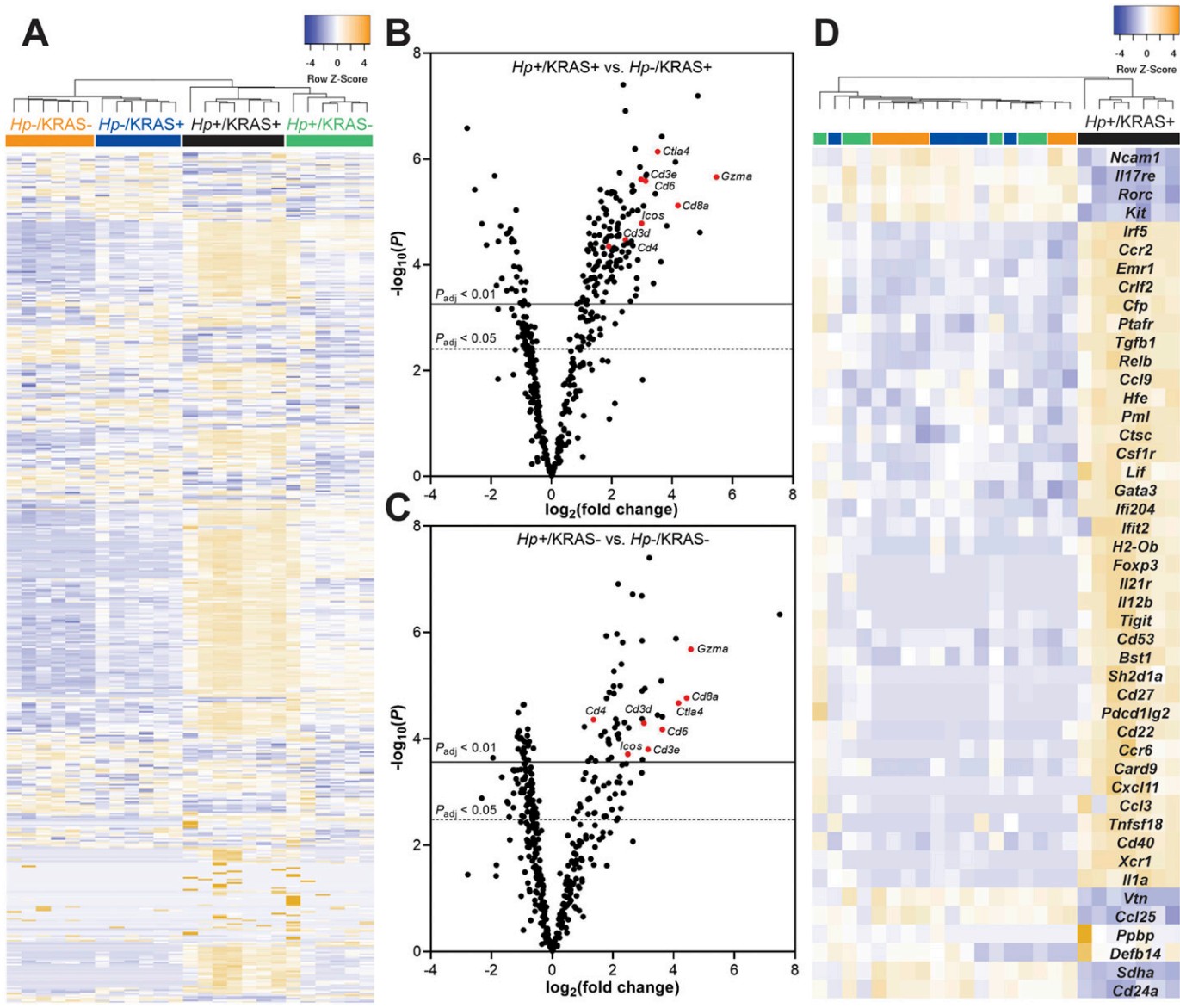

**Figure 4. A unique inflammatory gene signature exists in *Hp+*/KRAS+ mice at 12 wk.**

RNA was extracted from stomach sections from *Hp+/−*, KRAS+/− mice at 12 wk and immune-related gene expression was detected with the NanoString nCounter Mouse Immunology Panel. **(A)** Expression of 501 genes is shown. Colored bars denote different treatment groups. **(B, C)** Volcano plots show the fold change and *P*-values of all genes in the panel, for *Hp+*/KRAS+ mice versus *Hp−*/KRAS+ mice (B) and *Hp+*/KRAS− mice versus *Hp−*/KRAS− mice (C). A subset of T cell-related genes is shown in red and labeled. Lines show genes meeting the threshold for significance after correction with the Benjamini–Yekutieli procedure. **(D)** Expression of 46 genes (see text) that were uniquely differentially expressed in *Hp+*/KRAS+ mice versus all other groups is shown. The dendrograms at the top of the heat maps were produced by hierarchical clustering of gene expression. Data comes from N = 1 NanoString experiment with n = 6–7 mice per group from N = 2 independent mouse experiments.

the M1 polarization marker MHC class II. *Hp+*/KRAS+ mice had similar numbers of F4/80⁺/MHC class II⁺ cells present, but significantly fewer F4/80⁺/CD163⁺ cells (Fig S6B), suggesting altered macrophage polarization; most CD163 signal was observed in the gland lumen, likely nonspecific staining due to mucus binding. These IHC experiments confirm our gene expression-based findings that inflammation in *Hp+*/KRAS+ mice is not only more severe than in *Hp−*/KRAS+ mice but is also altered in nature.

### *Hp* infection alters metaplasia marker expression

We wondered whether the changes in tissue histology observed in *Hp+*/KRAS+ mice (Fig 2) reflected changes to the nature of metaplasia in these mice. To detect differences in hyperplasia, metaplasia, and cell proliferation in corpus tissue from *Hp+*/KRAS+ versus *Hp−*/KRAS+ mice over time (Fig 6A–C), we used conjugated

Statistically significant comparisons are indicated by: *P < 0.05, **P < 0.01, ****P < 0.0001, Kruskal–Wallis test with Dunn's multiple test correction. CFU/g, colony-forming units per gram stomach tissue. In (D, E), open versus closed symbols distinguish between biological replicates.

**Figure 5. Multiplex immunohistochemistry demonstrates more T cells and fewer M2 macrophages in *Hp+*/KRAS+ mice.**

Corpus tissue from *Hp−*/KRAS+ and *Hp+*/KRAS+ mice obtained after 12 wk was assessed by multiplex immunohistochemistry. Images were acquired with the Vectra Polaris Quantitative Pathology Imaging System (Akoya Biosciences) with a 20× objective. Phenochart was used to select multispectral image tiles, which were spectrally unmixed with Akoya inForm software and analyzed in HALO (Indica Labs). Representative images are shown and white boxes denote regions shown at higher magnification. **(A)** CD3 suggests T cells, CD4 indicates helper T cells, and CD8α indicates cytotoxic T cells. **(B)** CD3 suggests T cells, FOXP3 indicates regulatory T cells, and PD-1 is a marker of exhausted T cells. **(C)** F4/80 suggests macrophages, MHC class II indicates M1 macrophages, CD163 indicates M2 macrophages, and arrowheads indicate

lectin from *Ulex europaeus* (UEA-I), which binds α-L-fucose, to detect foveolar (pit cell) hyperplasia; conjugated *Griffonia simplicifolia* lectin II (GS-II), which binds α- or β-linked N-acetyl-D-glucosamine, to detect mucous neck cells and SPEM cells; anti-CD44v10 (orthologous to human CD44v9, referred to herein as "CD44v") to detect SPEM cells (Radyk et al, 2018); anti-TFF3 and anti-MUC2 to detect IM (goblet) cells (Merchant, 2005) (verified by staining of mouse intestine as shown in Fig S7); and anti-KI-67 to detect proliferating cells. We assessed differences through quantitative and semi-quantitative analysis of three to five images per mouse (Fig 6D–H). No difference was observed in UEA-I staining among the treatment groups (Fig S8), suggesting that *Hp* infection did not impact foveolar hyperplasia development in this model. In *Hp*−/KRAS+ mice, GS-II staining was observed at the base of the glands at 6 wk, co-localizing with CD44v, demonstrating SPEM (Fig 6A, D, and E). As well, most mice had robust, cell-associated TFF3 staining (Fig 6A and F) and a low degree of MUC2 staining (Fig 6B and G), suggestive of early IM. At 12 wk, CD44v and GS-II staining was reduced, TFF3 staining remained robust, and the percentage of MUC2$^+$ glands increased, suggesting a transition from SPEM to IM in these mice, consistent with previous findings (Choi et al, 2016). In *Hp*+/KRAS+ mice, GS-II and MUC2 staining patterns were similar, with GS-II decreasing and MUC2 increasing between 6 and 12 wk (Fig 6A, B, D, and G). However, CD44v and TFF3 exhibited a different pattern. At 6 wk, *Hp*+/KRAS+ mice had greater CD44v staining and less TFF3 staining than *Hp*−/KRAS+ mice (Fig 6A, E, and F). By 12 wk, CD44v staining waned somewhat but remained higher than in *Hp*−/KRAS+ mice, and TFF3 staining further diminished compared with *Hp*−/KRAS+ mice. Taken together, these results demonstrate that *Hp* infection alters the kinetics of metaplasia development in KRAS+ mice. Metaplasia marker expression was not detected in KRAS− mice (Fig S9). Of note, no differences in immunostaining patterns or quantification were observed in KRAS+ mice at 2 wk (Fig S10), suggesting that *Hp*-driven metaplastic changes take longer than 2 wk to become apparent. Finally, we assessed expression of phospho-ERK1/2, which is a downstream target of KRAS signaling (Choi et al, 2016), in +/− *Hp*, +/− Kras mice at 12 wk (Fig S11). Most signal in KRAS− mice with or without *Hp* infection was nonspecific, with only one or two positive cells per field of view. In contrast, KRAS+ mice had robust phospho-ERK1/2 signal that did not increase with *Hp* infection. Thus, the altered metaplasia observed in *Hp*+/KRAS+ mice is not due to enhanced KRAS activity.

We observed mitotic figures in KRAS+ mice at 12 wk (Fig 2F and H), suggesting increased cell division. Previously, patients with IM were found to have significantly increased cellular proliferation (assessed by KI-67 staining) in biopsy tissue compared with healthy controls and patients with chronic active gastritis (Erkan et al, 2012). Here, we found substantially more KI-67+ nuclei in corpus tissue of *Hp*+/KRAS+ mice than *Hp*−/KRAS+ mice at both 6 and 12 wk (Fig 6C and H). Most KI-67+ cells were found within the glandular epithelial compartment, not in the lamina propria, and interestingly, the localization of KI-67+ cells was altered in KRAS+ mice. In KRAS− mice,

proliferating cells were found in the middle of the glands, where gastric stem cells are found (Fig S9B). We observed that KI-67+ cells localized toward the base of the glands in *Hp*−/KRAS+ mice (Fig 6C). In *Hp*+/KRAS+ mice, KI-67 cells were abundant toward the base of the glands and higher up into the middle of the glands. In both groups of mice, some KI-67+ nuclei were found in GS-II+ cells at the base of the glands, suggesting proliferation of SPEM cells. However, in *Hp*+/KRAS+ mice, most KI-67+ nuclei were found above GS-II+ cells, suggesting proliferation of additional cell types beyond those with a SPEM phenotype.

### *Hp* infection increases dysplasia and cancer-associated gene expression

Overexpression of the calcium signal transducer TROP2 has been implicated in a variety of cancers (Shvartsur & Bonavida, 2015), including gastric cancer, where it is associated with worse outcomes (Muhlmann et al, 2009). Notably, TROP2 expression was recently identified as a strong indicator of the transition from incomplete IM to gastric dysplasia in *Mist1-Kras* mice and in human samples (Riera et al, 2020). We observed TROP2+ corpus glands by immunofluorescence microscopy at 6 and 12 wk (Figs 7A and S12). Quantitation using collagen VI as a gland segmentation marker revealed that *Hp*−/KRAS+ mice had TROP2 expression in 0–3.6% of glands at 6 wk, and 0–4.9% of glands at 12 wk (Figs 7B and S12). *Hp*+/KRAS+ mice had similar TROP2 expression at 6 wk (0.5–2.5% of glands), but at 12 wk had significantly more TROP2+ glands (1.5–9.1%, *P* < 0.05). Thus, the addition of *Hp* significantly increased the percentage of TROP2+ glands in the corpus at 12 wk. In all mice, most regions of TROP2 staining co-localized with KI-67 staining, suggesting proliferation of dysplastic glands. Only a few TROP2+ regions did not harbor KI-67+ cells (Fig 7A, asterisk). However, the association between TROP2 and KI-67 was greatest in *Hp*+/KRAS+ mice at 12 wk, where a median of 86% of TROP2+ glands or gland fragments were KI-67+ (*P* < 0.01) (Fig 7C), suggesting that *Hp* infection increases the proliferation of dysplastic glands.

We mined our NanoString gene expression data (Fig 4A) and identified 49 genes implicated in the development of gastrointestinal cancers (Alpizar–Alpizar et al, 2012; Lee et al, 2012; Jin et al, 2015; Huang et al, 2018; Min et al, 2019; Molina-Castro et al, 2020). Hierarchical clustering analysis showed that as with the overall panel, *Hp* infection status had the greatest impact on expression of this subset of genes at 12 wk, and that active KRAS expression also impacted gene expression, although to a lesser extent than *Hp* infection status did (Fig 7D). However, some genes, such as *Jak2*, *Notch2*, and *Runx1*, were up-regulated in KRAS+ mice regardless of infection status. Finally, metaplastic and dysplastic organoids generated from *Hp*−/KRAS+ mice at 12 and 16 wk after active KRAS induction, respectively, were previously found to have unique phenotypes and gene expression signatures (Min et al, 2019). Seven of these genes were found in our panel (Fig 7D, denoted with #). Expression of the metaplasia-associated gene *Clu* was strongly up-regulated in KRAS+ mice, but the metaplasia-associated gene *Ly86* was only strongly expressed in *Hp*+/KRAS+ mice. Of the dysplasia-

---

F4/80$^+$/CD163$^+$ cells. Scale bars for low magnification ("merge") images, 100 *µm*; for high magnification images, 25 *µm*. Data are from N = 1 multiplex immunohistochemistry experiment, with n = 7 mice per group from N = 2 independent mouse experiments. **(B, C)** Asterisks denote examples of nonspecific staining as determined by luminal location, lack of DAPI signal and lack of co-localization with relevant markers (B, CD3, and C, F4/80).

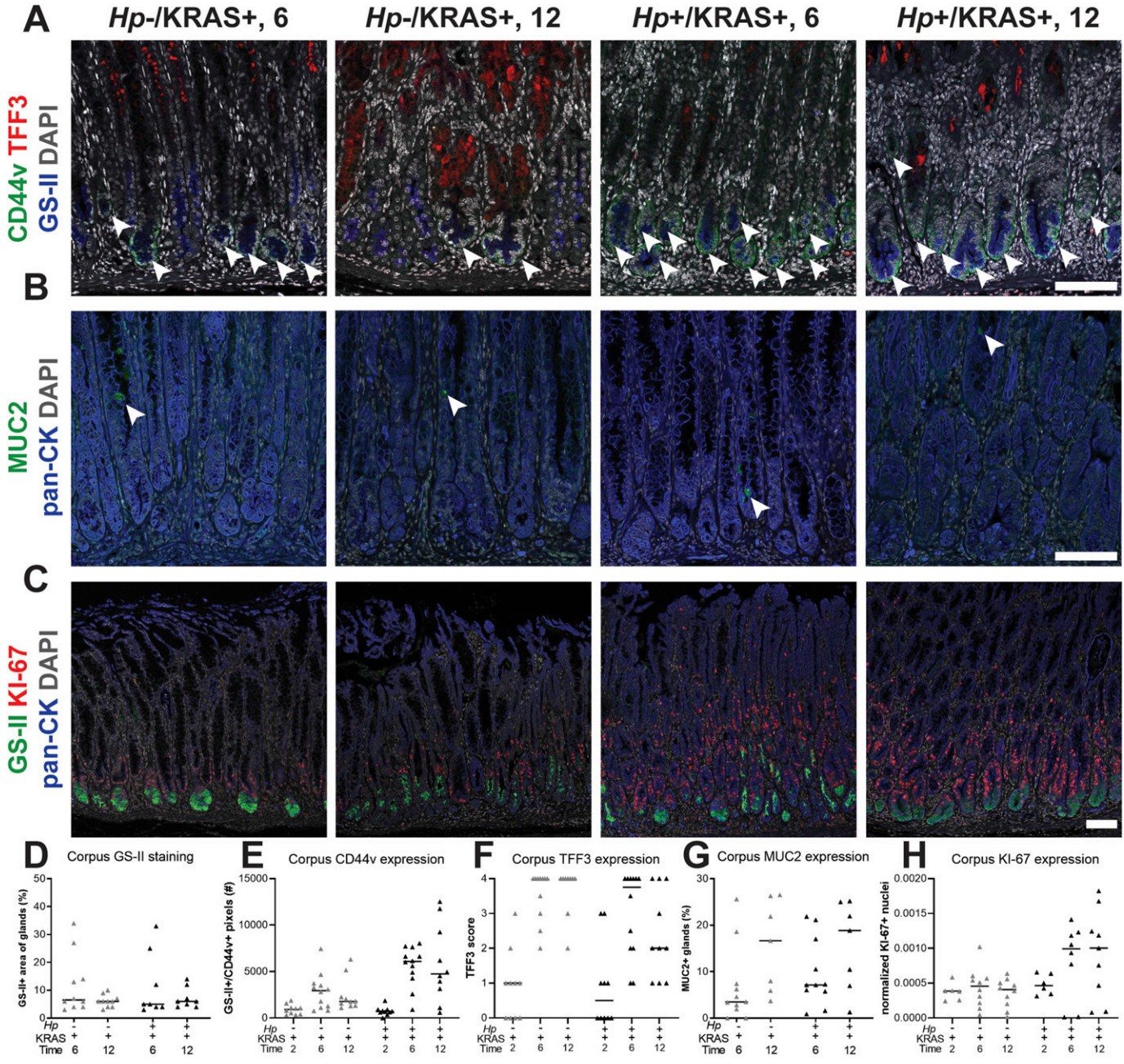

**Figure 6. The kinetics and molecular nature of metaplasia development are altered in *Hp+*/KRAS+ mice.**

Corpus tissue from *Hp−*/KRAS+ and *Hp+*/KRAS+ mice obtained after 2, 6, or 12 wk (N = 2 independent mouse experiments per time point and n = 6–12 mice per group) was assessed for metaplasia via immunofluorescence microscopy. **(A, B, C)** Images were taken with a Zeiss LSM 780 confocal microscope using 10× and 20× objectives and analyzed with Zen and Fiji software. Representative images are shown and scale bars are 100 μm. **(A)** Stomachs were stained with antibodies against CD44v (green; arrowheads) and TFF3 (red), the lectin GS-II (blue), and DAPI (grey) in N = 3 staining experiments. **(B)** Stomachs were stained with antibodies against MUC2 (green; arrowheads) and pan-cytokeratin (blue) and DAPI (grey) in N = 2 staining experiments. **(C)** Stomachs were stained with antibodies against KI-67 (red) and pan-cytokeratin (blue), the lectin GS-II (green) and DAPI (grey) in N = 3 experiments. **(D, E, F, G, H)** Three to five representative images per mouse were quantitatively or semi-quantitatively assessed and the median value for each mouse is plotted. Bars on the graphs indicate the median value for each mouse group. **(D)** The percentage of cytokeratin-positive epithelial tissue that was dual-positive for GS-II staining was detected. **(E)** The number of GS-II$^+$/CD44v$^+$ pixels per image was quantified. **(F)** TFF3 staining was semi-quantitatively scored in a blinded fashion. Nonspecific staining was not included in the score (see Fig S9). **(G)** The percentage of MUC2$^+$ glands was determined by counting. **(H)** KI-67$^+$/DAPI$^+$ nuclei were enumerated and normalized to the DAPI content (total number of DAPI$^+$ pixels) of each image.

associated genes, *Tubb5* was elevated in *Hp+* mice, *Gapdh* was elevated in KRAS+ mice, and *Eef1g* was not differentially expressed among treatment groups. Finally, *Cd44* and *Tgfb1* were previously found in both metaplastic and dysplastic organoids (Min et al, 2019)

and in our study were strongly elevated in *Hp+*/KRAS+ mice at 12 wk relative to the other mouse groups. Thus, gene expression in *Hp+*/ KRAS+ whole stomachs at 12 wk is distinct from *Hp−*/KRAS+ organoids generated at either 12 or 16 wk, further supporting the

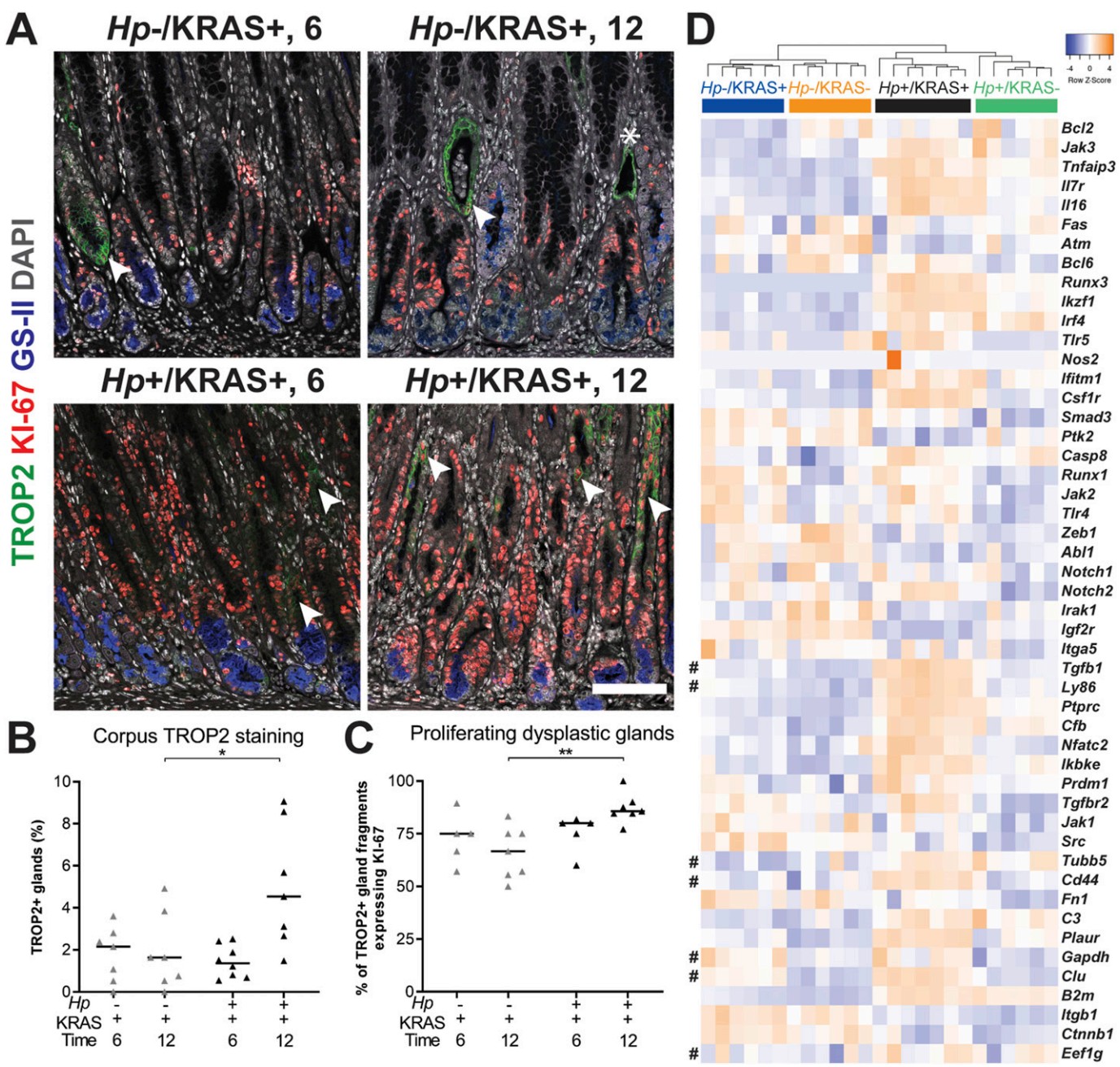

**Figure 7. *Hp* infection increases dysplasia and cancer gene expression in KRAS+ mice.**
**(A, B, C, D)** Stomachs from *Hp*−/KRAS+ and *Hp*+/KRAS+ mice were assessed through immunofluorescence microscopy (A, B, C) and gene expression analysis (D). **(A, B, C)** Corpus tissue from *Hp*−/KRAS+ and *Hp*+/KRAS+ mice obtained after 6 or 12 wk (N = 2 independent mouse experiments per time point and n = 5–8 mice per group) was stained with antibodies against TROP2 (green) and KI-67 (red), the lectin GS-II (blue), and DAPI (grey). **(A)** Images were taken with a Zeiss LSM 780 confocal microscope using a 20× objective and analyzed with Zen and Fiji software. Representative images are shown, arrows show TROP2⁺/KI-67⁺ gland fragments, and the asterisk shows a TROP2⁺/KI-67⁻ gland fragment. Scale bar, 100 μm. **(B)** TROP2⁺ gland fragments were enumerated as a percentage of total gland fragments detected. **(C)** TROP2⁺ glands or gland fragments were assessed for KI-67 staining and the percentage of dual-positive gland fragments is shown. Statistically significant comparisons are indicated by: *P < 0.05, **P < 0.01, Kruskal–Wallis test with Dunn's correction. **(D)** The expression of gastric cancer-associated genes discovered through literature searching is shown. RNA was extracted from stomach sections from *Hp*+/−, KRAS+/− mice at 12 wk and gene expression was detected with the NanoString nCounter Mouse Immunology Panel. The dendrogram was produced by hierarchical clustering of gene expression and colored bars denote different treatment groups. Data come from N = 1 NanoString experiment with n = 6–7 mice per group from N = 2 independent mouse experiments. # denotes genes expressed in *Mist1-Kras* organoids (Min et al, 2019).

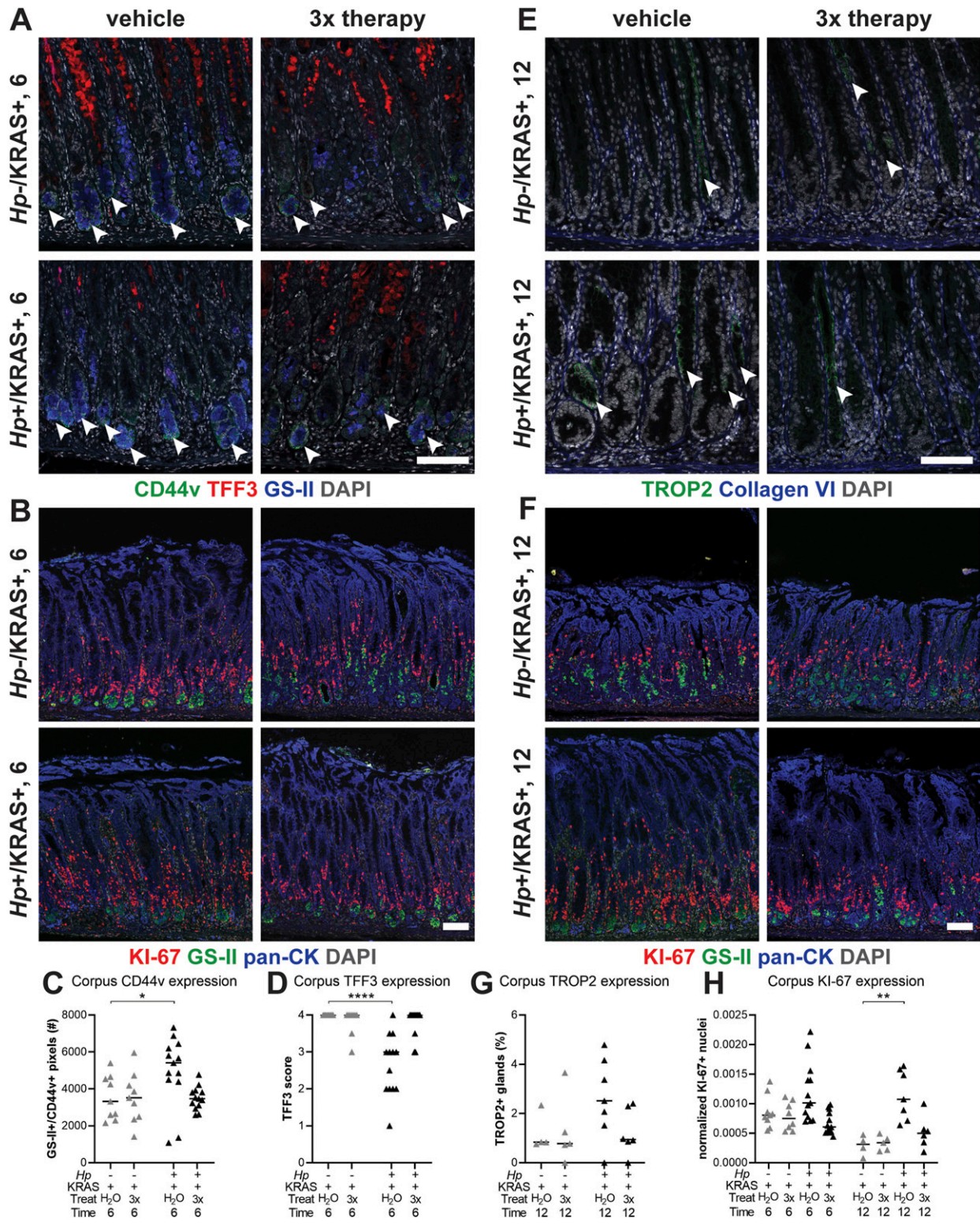

**Figure 8. Antibiotic eradication prevents *Hp*-associated changes in *Hp+*/KRAS+ mice.**

First, mice were infected with *Hp* or mock-infected, then active KRAS was induced. **(A, B, C, D, E, F, G)** Starting at 2 wk (A, B, C, D) or 6 wk (E, F, G), mice received 2 wk of daily antibiotic therapy (triple therapy of tetracycline, metronidazole, and bismuth, "3×") or vehicle control ("H₂O") Mice were euthanized after 6 (A, B, C, D) or 12 (E, F, G) wk and immunofluorescence microscopy was used to assess tissue phenotypes. **(A)** Stomachs were stained with antibodies against CD44v (green, arrowheads) and TFF3 (red), the lectin GS-II (blue), and DAPI (grey) in N = 1 staining experiment. **(E)** Stomachs were stained with antibodies against TROP2 (green, arrowheads) and collagen VI (blue) and DAPI (grey) in N = 1 staining experiment. **(B, F)** Stomachs were stained with antibodies against KI-67 (red) and pan-cytokeratin (blue), the lectin GS-II (green) and DAPI

hypothesis that concomitant *Hp* infection and active KRAS expression results in a unique gastric environment that is distinct from either *Hp*−/KRAS+ or *Hp*+/KRAS− mice.

### Sustained *Hp* infection is necessary to elicit changes to metaplasia, dysplasia and cell proliferation

Finally, we tested the impact of antibiotic eradication of *Hp* in two lines of experiments (Figs 8 and S13). First mice were infected with *Hp* or mock-infected, and active KRAS was induced. Starting at 2 wk after active KRAS induction, mice received 2 wk of "triple therapy" of tetracycline, metronidazole and bismuth (Arnold et al, 2011) or vehicle (water) as a control, and were euthanized at 6 wk (Fig 8A and B). Notably, *Hp*+/KRAS+ mice that received triple therapy had low CD44v (Fig 8A and C) and high TFF3 (Fig 8A and D) expression, similar to *Hp*−/KRAS+ mice (whether untreated as in Fig 6E and F, or treated with triple therapy or water). Thus, eradication of *Hp* early in the course of infection prevents the altered course of metaplasia seen at 6 wk. Similarly, we tested the effects of triple therapy after 6 wk (a time point where significant *Hp*-dependent changes to the tissue are already evident) with euthanasia at 12 wk (Fig 8E and F). *Hp*+/KRAS+ mice that received triple therapy had reduced TROP2+ glands at 12 wk (Fig 8E and G), suggesting that sustained *Hp* presence is necessary to accelerate dysplasia in this model. Finally, we found that at both time points antibiotic-treated *Hp*+/KRAS+ mice had reduced KI-67 staining (Fig 8B, F, and H), demonstrating that sustained *Hp* presence is also required for the hyperproliferation phenotype in *Hp*+/KRAS+ mice.

## Discussion

In this study we examined the effect of *Hp* infection in stomachs expressing active KRAS (Table 1). Up to 40% of human gastric cancers have genetic signatures of RAS activity (Deng et al, 2012; Cancer Genome Atlas Research Network, 2014). Activation of RAS and/or other oncogenic pathways in humans could be a consequence of *Hp*-driven inflammation. In our model, KRAS activation serves as a tool to model the consequences of oncogenic inflammation caused by *Hp* infection. Because active KRAS alone is sufficient to cause gastric preneoplastic progression (Choi et al, 2016), it might be expected that *Hp* infection would have no impact on KRAS-driven phenotypes. However, we found that *Hp* infection did influence preneoplastic progression in this model. *Hp* infection in KRAS-expressing mice led to more severe inflammation, an altered trajectory of metaplasia, substantial cell proliferation, and increased dysplasia compared with active KRAS alone. In addition, eradication of *Hp* with antibiotics prevented these tissue changes, in accordance with a major long-term study of *Hp* eradication in

Colombian adults with precancerous lesions, which showed that continuous *Hp* presence was significantly associated with disease progression (Mera et al, 2018). Thus, our study supports the hypothesis that sustained *Hp* infection can impact the molecular course of cancer development, beyond just initiating chronic inflammation.

Different mouse models exhibit different clinical features of preneoplastic progression. For example, *Helicobacter* infection alone causes SPEM but does not cause foveolar hyperplasia or IM in C57BL/6 mice, whereas uninfected *Mist1-Kras* mice exhibit all of these tissue states after active KRAS induction (Petersen et al, 2017). Here, we found that the combination of sustained *Hp* infection and active KRAS expression has a unique impact on the development of gastric metaplasia that is not observed with either individual parameter. Compared with *Hp*−/KRAS+ mice, *Hp*+/ KRAS+ mice had no difference in foveolar hyperplasia or expression of the IM marker MUC2, but had decreased expression of the IM marker TFF3 and increased expression of the SPEM marker CD44v. Further work is needed to determine whether these changes in metaplasia marker expression may reflect increased SPEM versus a process similar to incomplete IM. To our knowledge, of the various mouse models of gastric corpus preneoplastic progression, only *Mist1-Kras* mice exhibit true IM (indicated by TFF3[+] and MUC2[+] glands) in 100% of mice; most other models exhibit SPEM with or without intestinalizing characteristics (Petersen et al, 2017). The finding that *Hp* infection altered TFF3 expression in *Mist1-Kras* mice is therefore quite striking. TFF3 expression was moderate in both treatment groups at 2 wk, and it is not yet known whether TFF3 expression may have peaked in *Hp*+/KRAS+ mice at an intermediate time point, such as 4 wk, or was never as strongly expressed as in *Hp*−/KRAS+ mice. Notably, several human studies have reported that the association of SPEM with gastric adenocarcinoma is equal to or even greater than that of IM (Schmidt et al, 1999; Yamaguchi et al, 2002; Halldorsdottir et al, 2003), leading to questions in the field regarding the trajectory of metaplasia development before gastric cancer onset. Our findings suggest that the trajectory of metaplasia could differ depending on whether or not *Hp* remains present in the stomach throughout preneoplastic progression.

Sustained *Hp* infection also caused a striking increase in cell proliferation as indicated by KI-67 staining, and increased TROP2 staining at 12 wk. TROP2 expression was lower in our mice than what was previously reported (Riera et al, 2020), which may reflect differences in animal housing conditions, different methods of tissue fixation and processing, or components of the microbiome (although antibiotic perturbation in *Hp*−/KRAS+ mice had no effect on expression of TROP2 or metaplasia markers). Nonetheless, within our controlled experiments, TROP2 staining was greatest in *Hp*+/ KRAS+ mice, suggesting that infection accelerates the onset of

---

(grey) in N = 2 staining experiments. **(C, D, G, H)** Three to five representative images per mouse were quantitatively or semi-quantitatively assessed and the median value for each mouse is plotted. Bars on the graphs indicate the median value for each mouse group. **(C)** The number of GS-II[+]/CD44v[+] pixels per image was quantified. **(D)** TFF3 staining was semi-quantitatively scored in a blinded fashion. Nonspecific staining was not included in the score (see Fig S9). **(G)** The percentage of TROP2[+] gland fragments was determined by collagen VI staining to detect individual gland fragments. **(H)** KI-67[+]/DAPI[+] nuclei were enumerated and normalized to the DAPI content (total number of DAPI[+] pixels) of each image. Statistically significant comparisons are indicated by: *$P < 0.05$, **$P < 0.01$, ****$P < 0.0001$, Kruskal–Wallis test with Dunn's correction. Representative images are shown. Scale bars, 100 $\mu$m.

**Table 1.** Summary of differences between *Hp−*/KRAS+ and *Hp+*/KRAS+ mice.

| Parameter | *Hp−*/KRAS+ | *Hp+*/KRAS+ | Relevance to humans |
|---|---|---|---|
| Tissue histology | Inflammation, loss of parietal cells, surface epithelial cell hyperplasia, tortuous glands. Median HAI is 13 at 6 wk and 11 at 12 wk | Worsening of these parameters. Median HAI is 19 at 6 wk and 20.5 at 12 wk | Inflammation, parietal cell loss and metaplasia are hallmarks of gastric preoplastic progression (Kuipers, 1999; Kusters et al, 2006) |
| Immune gene expression | Less up-regulation of T cell and macrophage-related genes | Up-regulation of genes related to T cells and T-cell exhaustion, macrophages and gastric cancer; unique inflammatory gene signature | Many genes up-regulated in *Hp+*/KRAS+ mice were shown to be up-regulated in human gastrointestinal cancers (Alpizar-Alpizar et al, 2012; Lee et al, 2012; Jin et al, 2015; Huang et al, 2018; Molina-Castro et al, 2020) |
| T cells | Moderate T cells, predominantly CD4$^+$. T cells mostly restricted to the base of the glands | Many T cells, predominantly CD4$^+$, including FOXP3$^+$. T cells extend throughout the glands | *Hp* infection leads to recruitment and activation of T cells (Akhiani et al, 2002; Sayi et al, 2009; Velin et al, 2009), which may lead to mucosal damage and accumulation of mutations (Shi et al, 2010)(Stoicov et al, 2009). Exhausted T cells may be targeted by immune checkpoint inhibitors (Figueroa-Protti et al, 2019) |
| Macrophages | F4/80$^+$/CD163$^+$ (M2) cells are present, with a few F4/80$^+$/MHC class II$^+$ (M1) as well | Some F4/80$^+$/MHC class II$^+$ (M1) cells are present, with very few F4/80$^+$/CD163$^+$ (M2) | M2 macrophages are found in human SPEM and IM (Petersen et al, 2014). Macrophages can infiltrate into gastric tumors and are associated with worse surgical outcomes (Ishigami et al, 2003) |
| Metaplasia marker expression (6 wk) | Moderate CD44v9, high TFF3, low MUC2 | High CD44v9, moderate TFF3, low MUC2 | SPEM (CD44v10+) and IM (MUC2+, TFF3+) are both precursor lesions associated with human gastric adenocarcinoma (Schmidt et al, 1999; Yamaguchi et al, 2002; Halldorsdottir et al, 2003) |
| Metaplasia marker expression (12 wk) | Moderate CD44v9, high TFF3, low MUC2 | High CD44v9, low TFF3, low MUC2 | |
| Cell proliferation | Moderate numbers of KI-67$^+$ cells near the base of the glands | Far more of KI-67$^+$ cells, near the base and throughout the glands | Cell proliferation was greater in IM than in gastritis or healthy human stomachs (Erkan et al, 2012) |
| Dysplasia | Low TROP2 expression; most TROP2$^+$ glands are proliferative (KI-67$^+$) | Moderate TROP2 expression at 12 wk and almost all TROP2$^+$ glands are proliferative (KI-67$^+$) | TROP2 overexpression is associated with many cancers (Muhlmann et al, 2009; Shvartsur & Bonavida, 2015) and was recently shown to be a gastric dysplasia marker in humans (Riera et al, 2020) |
| Antibiotic therapy | No impact on metaplasia, dysplasia, or cell proliferation marker expression | *Hp* eradication prevents altered metaplasia, accelerated dysplasia and cell proliferation phenotypes; resembles *Hp–*/KRAS+ | *Hp* eradication prevented disease progression in patients with precancerous lesions (Mera et al, 2018). In cancer patients with *Hp*, eradication reduces metachronous cancer (Choi et al, 2018) |

HAI, histological activity index.

dysplasia. Although almost all of the TROP2$^+$ glands in *Hp+*/KRAS+ mice had co-localized KI-67 staining, demonstrating proliferation of dysplastic glands, there were also many KI-67$^+$ cells in TROP2$^−$ glands. Future studies will seek to elucidate the specific cell types that are proliferating in *Hp+*/KRAS+ mice. Despite the enhanced proliferation of dysplastic glands, *Hp+*/KRAS+ mice did not develop gastric tumors within 12 wk. One limitation of our study is that the *Mist1* promoter is expressed outside the stomach in secretory lineages, including the salivary glands. Approximately ~4 mo after active KRAS induction, *Mist1-Kras* mice require humane euthanasia because of salivary gland tumors. Thus, we cannot test whether *Hp* infection promotes even more severe phenotypes, such as tumor development, at later time points. Specifically targeting active KRAS

to the chief cells via other promoters could overcome this hurdle. However, *Hp+*/KRAS+ mice did have increased expression of genes known to be associated with gastrointestinal cancers. It may be that at least some of these genes are associated with gastric cancer simply because they reflect *Hp* infection, the biggest risk factor for gastric cancer development.

Our study does not define whether the combination of sustained *Hp* infection and active KRAS expression is additive (activation of parallel pathways) or synergistic (activation of overlapping pathways). The presence of a unique inflammatory gene signature and altered metaplasia marker expression in *Hp+*/KRAS+ mice could indicate that each perturbation activates distinct pathways. In addition, phospho-ERK1/2 staining was similar between *Hp+*/KRAS+ and *Hp–*/KRAS+ mice,

indicating that *Hp* infection does not further activate the MAP kinase pathway in KRAS+ mice. However, more studies are needed to address the overlap in the pathways downstream of *Hp* infection and active KRAS expression in this model. Interestingly, we found that a few *Hp*+/KRAS+ mice naturally cleared their infection, yet still had a high degree of immunopathology. A limitation of modeling *Hp* infection in mice is the inability to monitor bacterial burdens over time. It may be that the mice in question cleared their infection just before euthanasia, with no time for *Hp*-driven tissue changes to reverse. Alternatively, *Hp* infection may lead to a "point of no return," after which immunopathology develops even in the absence of *Hp*. This hypothesis has been used to explain the lack of detectable *Hp* in about half of human gastric tumors (Atherton & Blaser, 2009; Peleteiro et al, 2012). However, we found that antibiotic eradication of *Hp* after 6 wk prevented the accelerated dysplasia and hyperproliferation phenotypes at 12 wk. Thus, if a "point of no return" exists in this model, it must occur after 6 wk.

We note that no significant differences in metaplasia or dysplasia marker expression were observed between *Hp*−/KRAS+ mice and *Hp*+/KRAS+ mice at 2 wk, suggesting that the adaptive immune response to *Hp* infection may be what promotes the differences in metaplasia and dysplasia observed at later time points. This observation agrees with previous findings that T cells were necessary for *Helicobacter*-associated gastritis (Ernst et al, 1997; Eaton et al, 2001) and metaplasia development (Roth et al, 1999). The immune response seen in *Hp*+/KRAS+ mice at 6 and especially 12 wk far exceeded what was observed in either *Hp*+/KRAS− or *Hp*−/KRAS+ mice, and indeed is much greater than what is typically seen in *Hp* mouse models. Notably, this inflammation did not eradicate *Hp*: most *Hp*+/KRAS+ mice remained colonized at 12 wk. *Hp* cells were observed within the lumen of metaplastic glands, where they may be protected from direct immune cell interaction. In addition, *Hp* has multiple strategies to prevent immune-mediated clearance (Salama et al, 2013), including disruption of normal T cell functions by: triggering up-regulation of PD-L1, a T-cell inhibitory ligand that binds programmed cell death protein-1 (PD-1), on gastric epithelial cells, leading to T-cell exhaustion (Das et al, 2006); inducing anergy through promoting T-cell expression of the CTLA-4 co-receptor (Anderson et al, 2006); inhibiting T-cell proliferation and normal effector functions with the vacuolating cytotoxin VacA (Gebert et al, 2003); and the induction of tolerogenic dendritic cells, which promote the differentiation of naive T cells into immunosuppressive regulatory T cells (Salama et al, 2013). It remains unknown whether and to what extent *Hp* may disrupt T-cell functions in our model. *Hp*+/KRAS+ mice were unique in their up-regulation of *Foxp3* and had FOXP3[+] T cells at 12 wk, but these cells were not sufficient to limit immunopathology. As well, in *Hp*+/KRAS+ mice we observed strong transcriptional up-regulation of T-cell exhaustion-related genes, such as *Pdcd1* (PD-1), and *Ctla4*, implicated in T-cell anergy. By immunohistochemistry, we saw evidence of PD-1 expression in both *Hp*−/KRAS+ and *Hp*+/KRAS+ mice. Further studies are needed to characterize the exact nature of immune cell polarization differences between treatment groups; to confirm whether the T cells observed in *Hp*+/KRAS+ mice may be exhausted, anergic, or senescent; and to test whether immunosuppression or immunomodulation would be protective against *Hp*'s effects in the model. Nonetheless, it is clear that the combination of *Hp* infection and active KRAS expression leads to a potent and unique inflammatory state. Given that immunotherapy is still underused in gastric cancer (Zayac & Almhanna, 2020) and only a subset of patients benefit from such treatments (Figueroa-Protti et al, 2019), a better understanding of how active *Hp* infection may alter the immune microenvironment during gastric metaplasia and cancer development is urgently needed and may lead to new therapeutic strategies.

When *Hp* was first discovered to be a bacterial carcinogen, studies using tissue histology rarely detected *Hp* within gastric tumors. Such studies may have helped establish the belief that although *Hp* initiates the gastric cancer cascade, by the time gastric cancer is developed, *Hp* no longer matters—the so-called "hit-and-run" model. However, more sensitive methods detect *Hp* in about half of gastric tumors (Tang et al, 2005; Cristescu et al, 2015; Talarico et al, 2018), indicating that a large percentage of patients maintain active *Hp* infection throughout cancer development. Notably, *Hp* eradication combined with endoscopic resection of early gastric cancer significantly reduces metachronous gastric cancer (Choi et al, 2018). As well, a recent study of 135 *Hp*-seropositive subjects with IM found that patients with active *Hp* infection (determined by histology and/or sequencing) were significantly more likely to have somatic copy number alterations, and that patients with somatic copy number alterations were more likely to experience IM progression (Huang et al, 2018). Given these observations, there is an urgent need for preclinical models that identify unique features of gastric neoplasia with versus without concomitant *Hp* infection, both to understand the etiology of gastric cancer and to determine the impact of infection on different therapeutic approaches. We have shown here that *Hp* can significantly impact metaplasia and dysplasia development, which suggests that gastric preneoplastic progression can develop differently in the presence versus absence of *Hp*. Future studies will elucidate the molecular mechanism(s) through which *Hp* exerts its effects in this model, and test whether active *Hp* infection during metaplasia or cancer may represent a therapeutic vulnerability that could be targeted with immunotherapy.

# Materials and Methods

### Ethics statement

All authors had access to the study data and reviewed and approved the final manuscript. All mouse experiments were performed in accordance with the recommendations in the National Institutes of Health Guide for the Care and Use of Laboratory Animals. The Fred Hutchinson Cancer Research Center is fully accredited by the Association for Assessment and Accreditation of Laboratory Animal Care and complies with the United States Department of Agriculture, Public Health Service, Washington State, and local area animal welfare regulations. Experiments were approved by the Fred Hutch Institutional Animal Care and Use Committee, protocol number 1531.

### *H. pylori* strains and growth conditions

*H. pylori* strain PMSS1, which is also called 10700 and which is CagA[+] with an active type IV secretion system (Lee et al, 1997; Arnold et al, 2011), and derivatives were cultured at 37°C with 10% $CO_2$ and 10% $O_2$ in a trigas incubator (MCO-19M; Sanyo). Cells were grown on solid

media containing 4% Columbia agar (BD Biosciences), 5% defibrinated horse blood (HemoStat Laboratories) 0.2% β-cyclodextrin (Acros Organics), 10 μg/ml vancomycin (Sigma-Aldrich), 5 μg/ml cefsulodin (Sigma-Aldrich), 2.5 U/ml polymyxin B (Sigma-Aldrich), 5 μg/ml trimethoprim (Sigma-Aldrich), and 8 μg/ml amphotericin B (Sigma-Aldrich). For mouse infections, bacteria grown on horse blood plates were used to inoculate liquid media (BB10) containing 90% (vol/vol) Brucella broth (BD Biosciences) and 10% fetal bovine serum (Gibco), which was cultured shaking at 200 rpm overnight in a MaxQ 2000 orbital shaker (Thermo Fisher Scientific) and grown to an optical density at 600 nm of 0.4–0.6 (mid-log phase), from which an inoculum of ~5 × $10^7$ Hp cells per 100 μl BB10 was prepared. To determine Hp titers in the stomach, harvested tissues were weighed, homogenized, serially diluted, and plated on the solid media described above, with the addition of bacitracin (200 μg/ml; Acros Organics) to prevent growth of the stomach microbiota.

### Mist1-Kras mouse model

A breeding pair of Mist1-CreERT2 Tg/+; LSL-K-Ras (G12D) Tg/+ ("*Mist1-Kras*") mice on the C57BL/6 background, described previously (Choi et al, 2016), was obtained from Vanderbilt University (E Choi and JR Goldenring) and used to establish a colony at Fred Hutchinson Cancer Research Center. Mice were housed two to five per cage, with cages docked in HEPA-filtered ventilation racks that provide airflow control on a 12 h light/dark cycle, and had access to chow (LabDiet) and water ad libitum. At weaning, ear punches were collected and used for genotyping as previously described (Choi et al, 2016) with the following primers: Mist1 WT F: CCAAGATCGAGACCCTCACG; Mist1 WT R: ACACACACAGCCCTTAGCTC Mist1 Cre F: ACCGTCAGTACGTGAGATATCTT; Mist1 Cre R: CCTGAAGATGT TCGCGATTATCT; active KRAS F: TCTCTGCAGTTGTTGGCTCCAAC; active KRAS R: GCCTGAAGAACGAGATCAGCAGCC. Healthy 8- to 16-wk-old male and female mice (randomly allocated to treatment groups) were infected with 5 × $10^7$ mid-log culture Hp cells in 100 μl of BB10, or mock-infected with 100 μl of BB10, via oral gavage. To induce active (oncogenic) KRAS expression, mice received three subcutaneous doses of 5 mg of tamoxifen (Sigma-Aldrich) in corn oil (Sigma-Aldrich) over 3 d, or were sham-induced with corn oil, starting 1 d after Hp or mock infection. For antibiotic eradication, mice received "triple therapy" of 4.5 mg/ml metronidazole, 10 mg/ml tetracycline hydrochloride and 1.2 mg/ml bismuth subcitrate, or vehicle (water), by oral gavage (Arnold et al, 2011). Mice received six doses in 7 d, 2 wk in a row. Mice were humanely euthanized by $CO_2$ inhalation followed by cervical dislocation 2, 6, or 12 wk after infection and transgene induction. Stomachs were aseptically harvested and most of the forestomach (non-glandular region) was discarded, leaving only the squamocolumnar junction between the forestomach and glandular epithelium. Approximately one-third of the stomach was homogenized and plated to enumerate Hp. The remaining approximately two-thirds of the stomach was fixed in 10% neutral-buffered formalin phosphate (Thermo Fisher Scientific), then embedded in paraffin and cut into 4-μm sections on positively charged slides. Hp+/KRAS+ and Hp−/KRAS mice did not exhibit overall health differences; body weights and behaviors were similar at time of euthanasia.

## Histology

Stomach sections were stained with hematoxylin and eosin (H&E). A veterinary pathologist (AL Koehne) scored the slides in a blinded fashion according to criteria adapted from Rogers (2012). Corpus tissue was evaluated for inflammation, epithelial defects, oxyntic atrophy, hyperplasia (tissue thickness), hyalinosis, pseudopyloric metaplasia, mucous metaplasia, and dysplasia. The sums of the individual scores for each criterion were summed to generate an HAI score. Scoring criteria are described below. HAI was not correlated with sex or with age of mice at euthanasia.

### Inflammation
Multifocal aggregates of inflammatory cells merit a score of 1. As the aggregates coalesce across multiple high-power fields (40× objective), the score increases to 2. Sheets of inflammatory cells and/or lymphoid follicles in the mucosa or submucosa receive a score of 3. Florid inflammation that extends into or through the muscularis and/or adventitia is a score of 4.

### Epithelial defects
A tattered epithelium with occasional dilated glands is a score of 1. As the epithelium becomes attenuated and ectatic glands become more numerous, the score increases to 2. Inapparent epithelial lining of the surface with few recognizable gastric pits are given a score of 3. Score 4 is reserved for mucosal erosions.

### Oxyntic atrophy
The oxyntic mucosa is defined by the presence of chief and parietal cells. Loss of up to half of the chief cells merit a score of 1. In instances with near complete loss of chief cells and minimal loss of parietal cells, a score 2 is assigned. The absence of chief cells with half the expected number of parietal cells is given a score of 3. Score 4 signifies near total loss of both chief and parietal cells.

### Surface epithelial hyperplasia
This score indicates elongation of the gastric gland due to increased numbers of surface (foveolar) and/or antral-type epithelial cells. Relative to the expected length of a normal gastric pit, a score of 1 indicates a 50% increase in length. A score of 2 is twice the expected length, a score of 3 is three times the expected length, and a score of 4 is four times the expected length.

### Hyalinosis
This mouse-specific gastritis lesion refers to the presence of brightly eosinophilic round or crystalline structures in the murine gastric surface epithelium. The presence of epithelial hyalinosis is given a score of 1, whereas absence of hyalinosis is a score 0.

### Pseudopyloric metaplasia
Pseudopyloric metaplasia is the loss of oxyntic mucosa and replacement with glands of a more antral phenotype. The score indicates the amount of replacement by antralized glands. Less than 25% replacement is a score of 1, 26–50% replacement is a score of 2, 51–75% replacement is a score of 3, and greater than 75% replacement is a score of 4.

**Life Science Alliance**

### Mucous metaplasia

This mouse-specific gastritis lesion is defined as the replacement of oxyntic cells with mucous producing cells that resemble Brunner's glands of the duodenum. The score is assigned based on the percentage of mucosa affected. A score 1 indicates less than 25% involvement, a score of 2 indicates 26–50% involvement, a score of 3 is 51–75% involvement, and a score of 4 means that greater than 75% of the mucosa is involved.

### Dysplasia

Dysplasia indicates a cellular abnormality of differentiation. In score 1 lesions, the glands are elongated with altered shapes, back-to-back forms, and asymmetrical cellular piling. In score 2, the dysplastic glands may coalesce with glandular ectasia, branching, infolding, and piling of cells. Gastric intraepithelial neoplasia is given a score of 3 and invasive carcinoma is a score of 4. The dysplasia score describes the most severe lesion(s) in each mouse.

### Gene expression analysis

RNA was extracted from five 4-$\mu$m FFPE stomach sections per mouse using the AllPrep DNA/RNA FFPE Kit (QIAGEN). To specifically investigate the impact of *Hp* on the host immune response, inflammatory gene expression was detected using the nCounter Mouse Immunology Panel (NanoString), which detects more than 550 mouse immune-related genes. Gene expression differences were detected using nSolver software (NanoString) and are given in Table S1. Volcano plots were constructed by taking the $\log_2$ of the fold change and the $-\log_{10}$ of the unadjusted *P*-value for each gene. The $P_{adjusted}$ lines show genes meeting the threshold for significance after correction with the Benjamini–Yekutieli procedure for controlling the false discovery rate. Hierarchical clustering was performed and heat maps were generated through HeatMapper (Babicki et al, 2016) using the average linkage method with Euclidian distance, with $\log_2$-transformed gene expression data. Clustering was applied to rows (genes) and columns (mice). To identify the unique gene signature in *Hp*+/KRAS+ mice, gene expression values were normalized to the geometric mean of the expression in *Hp*−/KRAS− mice, and all genes were identified for which the geometric mean of the fold change in *Hp*+/KRAS+ mice was >2 and in *Hp*+/KRAS− and *Hp*−/KRAS+ mice was <1.5, or the geometric mean of the fold change in *Hp*+/KRAS+ mice was <0.5 and in *Hp*+/KRAS− and *Hp*−/KRAS+ mice was >0.667.

### Multiplex immunohistochemistry for immune cell detection

Slides were baked for 60 min at 60°C and then dewaxed and stained on a Leica BOND RX system (Leica) using Leica BOND reagents for dewaxing (Dewax Solution), antigen retrieval/antibody stripping (Epitope Retrieval Solution 2), and rinsing (Bond Wash Solution). Antigen retrieval and antibody stripping steps were performed at 100°C with all other steps at ambient temperature. Endogenous peroxidase was blocked with 3% $H_2O_2$ for 5 min followed by protein blocking with 10% normal mouse immune serum diluted in TCT buffer (0.05M Tris, 0.15M NaCl, 0.25% casein, 0.1% Tween-20, 0.05% ProClin300, pH 7.6) for 10 min. Primary and secondary antibodies are given in Table S2. The first primary antibody (position 1) was applied for 60 min followed by the secondary antibody application for 10 min and application of the reactive Opal fluorophore (Akoya) for 10 min. A high-stringency wash was performed after the secondary and tertiary applications using high-salt TBST solution (0.05M Tris, 0.3M NaCl,

and 0.1% Tween-20, pH 7.2–7.6). Undiluted, species-specific Polymer HRP was used for all secondary applications, either Leica's PowerVision Poly-HRP anti-Rabbit Detection or ImmPress Goat anti-Rat IgG Polymer Detection Kit (Vector Labs) as indicated in Table S2. The primary and secondary antibodies were stripped with retrieval solution for 20 min before repeating the process with the second primary antibody (position 2) starting with a new application of 3% $H_2O_2$. The process was repeated until seven positions were completed. For the eighth position, following the secondary antibody application, Opal TSA-DIG was applied for 10 min, followed by the 20 min stripping step in retrieval solution and application of Opal 780 Fluor for 10 min with high stringency washes performed after the secondary, TSA-DIG, and Opal 780 Fluor applications. Slides were removed from the stainer and stained with DAPI for 5 min, rinsed for 5 min, and coverslipped with Prolong Gold Antifade reagent (Invitrogen/Life Technologies). Slides were cured overnight at room temperature, and then whole slide images were acquired on the Vectra Polaris Quantitative Pathology Imaging System (Akoya Biosciences). The entire tissue was selected for imaging using Phenochart and multispectral image tiles were acquired using the Polaris. Images were spectrally unmixed using Phenoptics inForm software and exported as multi-image TIF files, which were analyzed with HALO image analysis software (Indica Labs). DAPI was used to detect individual cells and then cells expressing each marker were automatically detected based on signal intensity, and reported as a percentage of DAPI-positive cells.

### Immunofluorescence microscopy of epithelial phenotypes

Stomach sections were prepared as described above. To validate the antibodies used to detect IM, the entire intestinal tract from duodenum to colon was removed from an untreated C57BL/6 mouse. The cecum was discarded and the unflushed intestinal tract was rolled as a "Swiss roll," fixed in 10% neutral-buffered formalin, paraffin-embedded, and sectioned. Tissue sections were deparaffinized with Histo-Clear solution (National Diagnostics) and rehydrated in decreasing concentrations of ethanol. Antigen retrieval was performed by boiling slides in 10 mM sodium citrate (Thermo Fisher Scientific) or Target Retrieval Solution (Agilent Dako) in a pressure cooker for 15 min. Slides were incubated with Protein Block, Serum Free (Agilent Dako) for 90 min at room temperature. Primary antibodies (Table S3) were diluted in Protein Block, Serum Free, or Antibody Diluent, Background Reducing (Agilent Dako), and applied to the slides overnight at 4°C. Secondary antibodies were diluted 1:500 in Protein Block, Serum Free and slides were incubated for 1 h at room temperature protected from light. Slides were mounted in ProLong Gold Antifade reagent with DAPI (Invitrogen) and allowed to cure for 24 h at room temperature before imaging. Slides were imaged on a Zeiss LSM 780 laser-scanning confocal microscope using Zen software (Zeiss) and three to five representative images of the corpus were taken.

### Quantitation of staining

Three to five representative images of corpus tissue per mouse were used for staining analysis, and the median value was reported for each mouse. Investigators were blinded to the treatment groups. Tissues were analyzed in Zen (ZEISS) and Fiji (Schindelin et al, 2012). KI-67, GS-II, CD44v10 (orthologous to human CD44v9 and referred to herein as "CD44v"), and TROP2 markers were quantified from fluorescently immunolabelled tissue sections by custom-made scripts developed in MATLAB 2019a.

Scripts can be found on Github at https://github.com/salama-lab/stomach-image-quantitation.

After background subtraction and denoising in each channel, positive pixels for DAPI, GS-II, CD44v, or TROP2 were identified by image binarization using the Otsu method and morphological filtering. When appropriate, individual glands were segmented using the cytokeratin signal, which is predominant in glandular structures, or using the complement image of the collagen VI signal, which is excluded from glandular structures. For GS-II quantification, the fractional area of cytokeratin staining positive for GS-II was recorded. For TROP2 quantification, the fractional area of each gland fragment identified by collagen VI labeling was recorded, and gland fragments with ≥10% TROP2-positive pixels were considered TROP2-positive. To identify GS-II and CD44v double-positive regions, the GS-II binary mask was first dilated by a few pixels because GS-II is cytoplasmic and CD44v is membrane-bound. The resulting number of overlapping pixels per image was then recorded. To assess KI-67 staining, individual KI-67-positive nuclei were identified using a watershed algorithm after distance transformation of the binarized signal, and then normalized by dividing by the total number of DAPI-positive pixels in the image.

TFF3, MUC2, and dual-positive TROP2/KI-67 staining were manually assessed. To assess TFF3 staining, images were scored manually using a semi-quantitative scale with the following criteria: 0 = no staining, 1 = 1–25% of glands are positive, 2 = 26–50% of glands are positive, 3 = 51–75% of glands are positive, 4 ≥ 75% of glands are positive for TFF3. Positive TFF3 signal manifests as moderately bright, cell-associated staining with goblet cell-like morphology. Overly bright staining without distinct goblet-like morphology, and/or staining within the gland lumen (not cell-associated), was observed near the top of the glands in $Hp-$/KRAS– (healthy control) mice and was considered false-positive staining. To assess MUC2 staining, glands were detected by cytokeratin staining, and $MUC2^+$ and $MUC2^-$ glands were manually counted. To assess gland fragments dual-positive for TROP2 and KI-67, regions of $TROP2^+$ staining that contained $KI-67^+$ nuclei were counted and expressed as a percentage of all $TROP2^+$ glands.

### Type IV secretion system activity

$Hp$ strain PMSS1 was recovered from infected mice after euthanasia by serial dilution plating, described above, and five or six individual colonies per mouse were expanded and frozen at –80°C. Colonies were then grown and used in co-culture experiments with AGS cells (from a human gastric adenocarcinoma cell line; ATCC CRL-1739) as previously described (Martinez et al, 2019). The input strain of PMSS1 (freezer stock) served as a positive control and a PMSS1Δ$cagE$ mutant (Martinez et al, 2019) served as a negative control. Infections were performed in triplicate and supernatants were collected after 24 h of co-culture. IL-8 was detected using a human IL-8 ELISA kit from BioLegend.

### Statistical analyses

Volcano plots and heat maps were constructed from data generated by nSolver software (NanoString). For NanoString analysis, $P_{adjusted}$ values were generated in nSolver with the Benjamini-Yekutieli procedure for controlling the false discovery rate. Other statistics were performed in GraphPad Prism v7.01. Comparisons of three or more groups were performed with the Kruskal–Wallis test followed by Dunn's multiple test correction. $P < 0.05$ was considered statistically significant. For histopathological evaluation of stomach sections and quantitation of staining, experimenters were blinded to the treatment groups.

## Data Availability

Gene expression data are found in Table S1.

## Supplementary Information

## Acknowledgements

The authors wish to acknowledge Savanh Chanthaphavong and Louis Kao for assistance with experimental histopathology and Alicia M Meyer for assistance with mouse experiments. This work was funded by an Innovation Grant from the Pathogen-Associated Malignancies Integrated Research Center at Fred Hutchinson Cancer Research Center and National Institutes of Health (NIH) R01 AI54423 (to NR Salama). Research was supported by the Cellular Imaging, Comparative Medicine, Genomics & Bioinformatics, and Research Pathology Shared Resources of the Fred Hutch/University of Washington Cancer Consortium (P30 CA015704). VP O'Brien is a Cancer Research Institute Irvington Fellow supported by the Cancer Research Institute, and was also supported by a Debbie's Dream Foundation—American Association for Cancer Research (AACR) Gastric Cancer Research Fellowship, in memory of Sally Mandel (18-40-41-OBRI). AE Rodriguez was supported by the Jacques Chiller Award from the Department of Microbiology, University of Washington. JR Goldenring was supported by Department of Veterans Affairs Merit Review Award IBX000930, NIH R01 DK101332, Department of Defense (DOD) CA160479, and a Cancer UK Grand Challenge Award. E Choi was supported by DOD CA160399, NIH R37 CA244970, and pilot funding from Vanderbilt Digestive Disease Research Center DK058404 and a Vanderbilt-Ingram Cancer Center GI SPORE P50CA236733.

### Author Contributions

VP O'Brien: conceptualization, formal analysis, funding acquisition, validation, investigation, visualization, methodology, and writing—original draft, review, and editing.

AL Koehne: investigation, methodology, and writing—review and editing.

J Dubrulle: software, formal analysis, methodology, and writing—review and editing.

AE Rodriguez: investigation and writing—review and editing.

CK Leverich: investigation and writing—review and editing.

VP Kong: investigation, methodology, and writing—review and editing.

JS Campbell: investigation and writing—review and editing.

RH Pierce: conceptualization, resources, and writing—review and editing.

JR Goldenring: conceptualization, resources, and writing—review and editing.

E Choi: conceptualization, resources, and writing—review and editing.

NR Salama: conceptualization, supervision, funding acquisition, and writing—review and editing.

**Conflict of Interest Statement**

The authors declare that they have no conflict of interest.

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
