## [Reviewer comments · Life Science Alliance]

Life Science Alliance

Sustained *Helicobacter pylori* infection accelerates gastric dysplasia in a mouse model

Valerie O'Brien, Amanda Koehne, Julien Dubrulle, Armando Rodriguez, Christina Leverich, Paul Kong, Jean Campbell, Robert Pierce, James Goldenring, Eunyoung Choi, and Nina Salama

DOI: <https://doi.org/10.26508/lsa.202000967>

Corresponding author: Nina Salama, Fred Hutchinson Cancer Research Center

Review Timeline:

Submission Date:	2020-11-19
Editorial Decision:	2020-11-20
Revision Received:	2020-11-23
Accepted:	2020-11-24

Scientific Editor: Shachi Bhatt

Transaction Report:

(Note: Please note that the manuscript was previously reviewed at another journal and the reports were taken into account in the decision-making process at Life Science Alliance. With the exception of the correction of typographical or spelling errors that could be a source of ambiguity, letters and reports are not edited. The original formatting of letters and referee reports may not be reflected in this compilation.)

November 20, 2020

RE: Life Science Alliance Manuscript #LSA-2020-00967-T

Dr. Nina Salama
Fred Hutchinson Cancer Research Center
Seattle

Dear Dr. Salama,

Thank you for submitting your revised manuscript entitled "Sustained *Helicobacter pylori* infection accelerates gastric dysplasia in a mouse model". We would be happy to publish your paper in Life Science Alliance (LSA). pending final revisions as noted below, and necessary to meet our formatting guidelines.

For a brief overview, the manuscript was reviewed at an external, non-Alliance journal, where it was rejected as the reviewers were concerned about conceptual advance. The authors then transferred the revised manuscript, reviewer comments and point-by-point rebuttal to LSA. LSA editors deemed the advance and mechanistic depth to be sufficient, and were satisfied with the revisions that the authors had already done in response to the concerns raised at the previous journal. The manuscript should be publishable at LSA pending the following minor edits,

- + please include the p-Erk1/2 staining results that were shown in response to Rev 2 pt 1, from the pbp rebuttal to the main manuscript. You can add them as supplemental data
- + please add a few lines of discussion on the additive vs. synergistic nature of relationship between HP infection and Kras changes in the manuscript text (response to Rev 2 pt 7)
- + please add the description of why the particular immunology panel was chosen (response to Rev 3 pt 2) to the main manuscript text

Along with the points listed above, please also attend to the following to make sure that the final file complies with LSA's formatting guidelines,

- please consult our Manuscript Preparation Guidelines <https://www.life-science-alliance.org/manuscript-prep> and put your manuscript sections in the correct order
- please add ORCID ID for corresponding author-you should have received instructions on how to do so
- please upload your main and supplementary figures as single files and add a separate section with your figure legends to your manuscript text
- please add the author contributions and a Summary Blurb / Alternate Abstract, to the system
- please add a conflict of interest statement to your manuscript
- please deposit large dataset in a publicly available dataset (<https://www.life-science-alliance.org/manuscript-prep#datadepot>)
- please specify the category of the manuscript as Research Article when submitting the revision

To upload the final version of your manuscript, please log in to your account:
<https://lsa.msubmit.net/cgi-bin/main.plex>

A. FINAL FILES:

B. MANUSCRIPT ORGANIZATION AND FORMATTING:

Thank you for your attention to these final processing requirements. Please revise and format the

manuscript and upload materials within 7 days.

Sincerely,

Shachi Bhatt, Ph.D.

Executive Editor

Life Science Alliance

<https://www.lsajournal.org/>

November 24, 2020

RE: Life Science Alliance Manuscript #LSA-2020-00967-TR

Dr. Nina Reda Salama
Fred Hutchinson Cancer Research Center
Human Biology
1100 Fairview Ave N
Mail Stop C3-168
Seattle, WA 98109

Dear Dr. Salama,

Thank you for submitting your Research Article entitled "Sustained *Helicobacter pylori* infection accelerates gastric dysplasia in a mouse model". It is a pleasure to let you know that your manuscript is now accepted for publication in Life Science Alliance. Congratulations on this interesting work.

DISTRIBUTION OF MATERIALS:

Again, congratulations on a very nice paper. I hope you found the review process to be constructive and are pleased with how the manuscript was handled editorially. We look forward to future exciting submissions from your lab.

Sincerely,

Shachi Bhatt, Ph.D.

Executive Editor

Life Science Alliance

<https://www.lsjournal.org/>
